# Research on the Adaptability of Typical Denoising Algorithms Based on ICESat-2 Data

Mengyun Kui [1,2,3], Yunna Xu [1,*], Jinliang Wang [1,2,3] and Feng Cheng [1,2,3]

[1] Faculty of Geography, Yunnan Normal University, Kunming 650500, China; kmengyun@163.com (M.K.); jlwang@ynnu.edu.cn (J.W.); 180009@ynnu.edu.cn (F.C.)

[2] Key Laboratory of Resources and Environmental Remote Sensing for Universities in Yunnan Kunming, Kunming 650500, China

[3] Center for Geospatial Information Engineering and Technology of Yunnan Province, Kunming 650500, China

* Correspondence: xuyunna_rs@163.com

**Abstract:** Photon-counting light detection and ranging (LiDAR) emits and receives weak photon signals, which are easily mixed with background noise caused by the sun, the atmosphere, etc., and is thus difficult to distinguish. Therefore, point-cloud denoising is a key step in point-cloud data processing of photon-counting LiDAR. To explore the adaptability of different denoising algorithms for photon-counting LiDAR data in different times and spaces, in this paper, NASA's official differential, regressive and Gaussian adaptive nearest neighbor (DRAGANN) algorithm; Herzfeld's radial basis function (RBF) denoising algorithm; and the density-based spatial clustering of applications with noise (DBSCAN) algorithm based on density clustering are used to denoise the ICESat-2 ATL03 photon point-cloud data. Airborne LiDAR data are used to verify the denoising accuracy, and then the adaptability of the three algorithms is discussed. The results show that the DRAGANN algorithm is suitable for data with moderate Fraction Vegetation Coverage (FVC) (45–75%) at night and is less affected by slope; therefore, it is not limited to terrain slope. The denoising accuracy of the RBF algorithm decreases with increasing FVC and decreases with increasing slope. It is suitable for data with low terrain slope (0~55°) and low FVC (0~220°), which is less affected by observation time; therefore, it is suitable for all-day data. The DBSCAN algorithm is suitable for data with moderate FVC (45~75%) at night, regardless of terrain slope. Unlike the DRAGANN algorithm, the DBSCAN algorithm is greatly affected by solar noise photons, but at night, its denoising accuracy is higher than that of the DRAGANN algorithm. The research results have certain reference significance for the subsequent processing and application of ICESat-2 data.

**Keywords:** ICESat-2; photon-counting LiDAR; point-cloud denoising algorithm; adaptability

## 1. Introduction

LiDAR is an active remote-sensing detection technology that has developed rapidly in recent years. It can quickly and directly obtain high-precision, three-dimensional point clouds of targets. It has strong penetration, strong anti-interference ability and high angle, distance and velocity resolutions. It has become an important means of three-dimensional Earth observation with high spatial and temporal resolution [1]. It plays an increasingly important role in environmental monitoring, digital cities, forestry surveys, global change, power inspection and other fields [2,3]. Most of the existing LiDAR systems use linear detection and full waveform sampling, and the large power consumption and volume weight lead to low detection efficiency. At the same time, the development and application of LiDAR are further limited by the low reflectivity of the target, the poor laser energy of the system and the poor sensitivity of the detector [4]. GEDI (Global Ecosystem Dynamics Investigation) and ICESat-2 (Ice, Cloud and land Elevation Satellite-2), developed by NASA, can effectively solve the above problems. GEDI is susceptible to waveform spreading, which allows the superposition of canopy and ground echoes, thus affecting the accurate

measurement of elevation and structural information [5]. ICESat-2 carries the Advanced Topographic Laser Altimeter System (ATLAS), which adopts the micropulse, multibeam photon-counting LIDAR technology, and has the characteristics of high re-frequency, high sensitivity, etc [6]. Compared with GEDI, ATLAS is able to fully acquire photon point-cloud data with higher density and smaller spot size, which is often applied to ice sheet height [7], sea ice thickness [8], lake level monitoring [9,10] and monitoring of the biomass of the Earth's forests [11,12].

The laser pulses emitted and received by the micropulse, photon-counting LiDAR are weak signals, and it is difficult to distinguish the return pulse signal from the background noise (mainly including solar radiation, atmospheric scattering sound and system noise) [13]. Therefore, the separation of signal and noise from photon-counting LiDAR point-cloud data is the premise and basis of ICESat-2 photon point-cloud data processing.

For the removal of ICESat-2 background noise, scholars have developed a series of denoising algorithms, which are mainly divided into supervised classification and unsupervised classification. Based on the idea of supervised classification, Lu [14] proposed a photon point-cloud denoising algorithm based on a convolutional neural network, which has achieved good denoising classification results in the case of prior information (bare land and forest). However, it is greatly affected by training samples, and the characteristic variables required under different terrain and surface coverage are different. For such algorithms, how to process the photon point-cloud data of more scenes based on the deep learning method when the prior information is unknown is a problem that needs further study. Unsupervised classification can be divided into the following three categories: (1) The first is a denoising algorithm based on a grid two-dimensional image; the profile photons are converted to two-dimensional image, and the noise is removed with image-processing techniques, such as the contour detection algorithm proposed by Awadallah M. [15], in which the algorithm principle is simple, but the point-cloud rasterization process causes information loss. (2) Then there is the denoising algorithm based on local statistical parameters; that is, by calculating the local statistics (distance, elevation, point density, feature vector, etc.) of each point of the photon point cloud, the global threshold is set by using its distribution characteristics (such as the histogram) to realize the classification of noise and signal [16–18]. However, the denoising effect of such algorithms depends on the selection of statistical parameter thresholds. The main algorithms include NASA's official ATL08 product, which uses the DRAGANN algorithm based on local density statistics, and Herzfeld's quadratic denoising algorithm for photonic point clouds based on the Gaussian radial basis function of local feature vectors (referred to as the RBF algorithm in the text) [19]. (3) The last category is denoising algorithms based on density clustering, denoising using the dense distribution of signal photons and the relatively sparse distribution of noise photons; such algorithms are sensitive to the input parameters and lack universality, such as Bayesian [20], DBSCAN [21] and OPTICS (Ordering Points To Identify the Clustering Structure) [22]. Among them, the DBSCAN algorithm is relatively simple in principle and one of the most commonly used means for photon denoising [23].

The existing studies mostly optimize the DRAGANN, RBF and DBSCAN algorithm parameter acquisition methods to make the above three algorithms have high applicability. For example, Huang et al. [24] established a quantitative relationship model between the number of neighborhood points P and the signal-to-noise ratio to derive the value of P in a specific scenario by improving the DRAGANN algorithm and optimized the Gaussian parameter, which not only has the adaptive search capability but also achieves the accurate segmentation of the signal and the noise. Li et al. [25] propose a push-and-sweep photon-counting LiDAR point-cloud filtering method by improving the RBF algorithm. Zhang et al. [26] optimized the key parameters of the DBSCAN algorithm according to the slope–noise relationship and changed the search neighborhood from the original circular neighborhood to an elliptical neighborhood, which improved the usability of the DBSCAN algorithm. However, most denoising algorithms are only for surface types under single spatio-temporal conditions and cannot meet the scenarios under complex surface coverage.

How to develop or improve photon point-cloud denoising algorithms that are adaptable to various scenarios is a question that requires further in-depth research.

At present, there are fewer studies on the adaptability of each denoising algorithm in different scenarios, so, in order to investigate the adaptability of the denoising algorithms for photon-counting LiDAR data under different spatial and temporal conditions, this paper selects three typical algorithms, DRAGANN, RBF and DBSCAN, to carry out denoising experiments on ICESat-2 data; analyzes the denoising effects of the three denoising algorithms under different FVC, slope and time conditions; uses airborne LiDAR point-cloud data to verify the denoising accuracy; and finally conducts a comprehensive analysis and comparison of the denoising effects.

## 2. Materials and Methods

### 2.1. Data Description and Experimental Area

2.1.1. ICESat-2 Data

ICESat-2 adopts micropulse multibeam photon-counting LIDAR technology; therefore, it does not need to record the number of collected photons and waveform information but collects only the returning photons with geographical location characteristics and obtains the geographical information of the actual area according to the distribution of the photons. It emits six laser pulses with different energies at a repetitive frequency of 10 kHz, divided into three groups along the track direction. Each group contains one strong and one weak signal, each group has a cross-track interval of approximately 3.3 km, and the cross-track interval within the group is approximately 90 m. The laser beam configuration is shown in Figure 1.

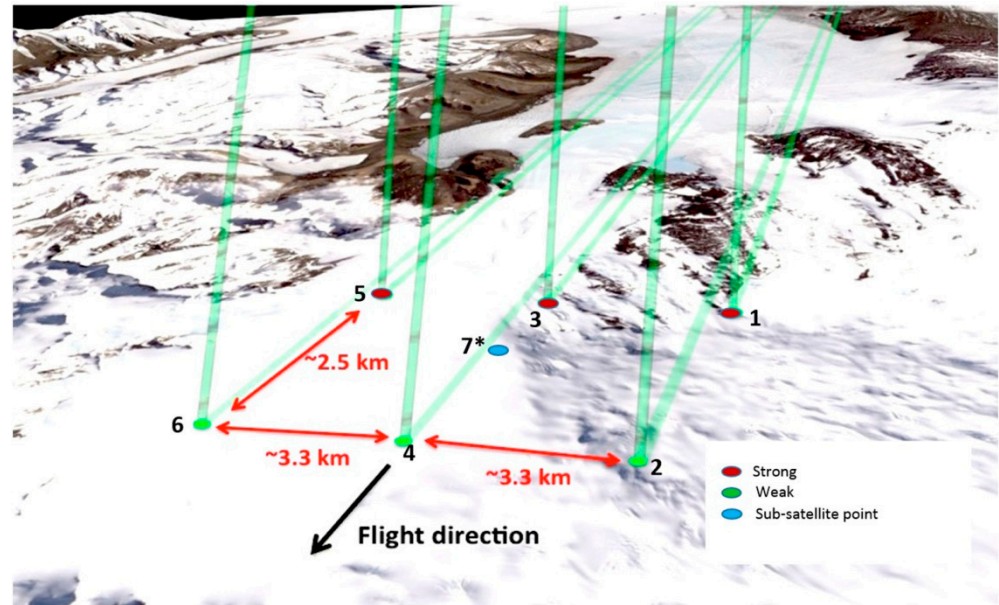

**Figure 1.** Schematic representation of the distribution of the 6 laser beams and (virtual) sub-satellite points (7*) of the ICESat-2 satellite. The laser beams are divided into three groups along the orbital direction, each containing a strong and a weak beam with an energy ratio of 4:1, which is used to adapt to the reflective properties of different surface types in order to provide better elevation measurement data [27] (Reprinted with permission from Ref. [27]. 2018, Elsevier Inc).

ICESat-2 provides 21 standard data products, ATL00-ATL21, classified as Level 0, Level 1, Level 2 and Level 3. ATL00 is a Level 0 product, which has raw telemetry data; ATL01 and ATL02 are Level 1 products, which have formatted and instrument-corrected telemetry data; ATL03 and ATL04 are Level 2 products; and ATL03 and ATL04 are Level 2 products. ATL03 has global geolocation photon data, providing time, latitude, longitude and ellipsoidal height for each photon event, and ATL04 is the backscatter correction file;

ATL05-ATL21 are Level 3 products, including land ice–surface height, sea-ice elevation, topography height and canopy height. Among them, this data set (ATL03) contains height above the WGS 84 ellipsoid (ITRF2014 reference frame), latitude, longitude and time for all photons downlinked by the ATLAS instrument on board the ICESat-2 observatory. Heights are corrected for several geophysical phenomena, such as effects of the atmosphere and solid earth deformation. The ATL03 is a single data source for all photonic data and ancillary information required for advanced ATLAS/ICESat-2 products. Therefore, in this paper, ATL03 is used to conduct experiments, and ICESat-2 data are available to the public in the conventional HDF5 data format, which can be downloaded for free from the National Snow and Ice Data Center (https://nsidc.org/data/atl03, accessed on 25 May 2023).

### 2.1.2. Airborne LiDAR Data and Experimental Area

The National Ecological Observatory Network (NEON) was established by the National Science Foundation (NSF) to collect high-quality standardized data on climate change and land-use change from 81 sites (47 terrestrial and 43 aquatic) across the United States to study important ecological and environmental issues, to predict trends in environmental change and to propose corresponding countermeasures [28]. In this paper, the airborne LiDAR data products provided by NEON were used as validation data and can be downloaded from a website (https://www.neonscience.org/data-collection/LiDAR, accessed on 10 January 2023).

To ensure the accuracy of the validation, airborne LiDAR data from five sites—Moab NEON (MOAB), Onaqui NEON (ONAQ), Bartlett Experimental Forest NEON (BART), Harvard Forest and Quabbin Watershed NEON (HARV) and Niwot Ridge NEON (NIWO)— were selected (Figure 2), and the observations were made in the same year and season as the ICESat-2 data. According to existing studies and grading criteria, a fractional vegetation cover (FVC) of 0–20% was designated as Class I, 20–45% as Class II, 45–75% as Class III and 75–100% as Class IV [29], and a topographic slope of 0–5° was designated as Class I, 5–15° as Class II, 15–25° as Class III and 25–60° as Class IV [30]. The difference between daytime and night-time solar noise photons is relatively large; therefore, the acquisition time was divided into two time periods: daytime and night-time. Table 1 shows the corresponding information of the ICESat-2 experimental data and airborne LiDAR data. Among them, Data1~ Data8 were mainly used to explore the FVC effect of the denoising effect of the three algorithms, and Data9~Data16 were mainly used to explore the slope effect of the denoising effect of the three algorithms.

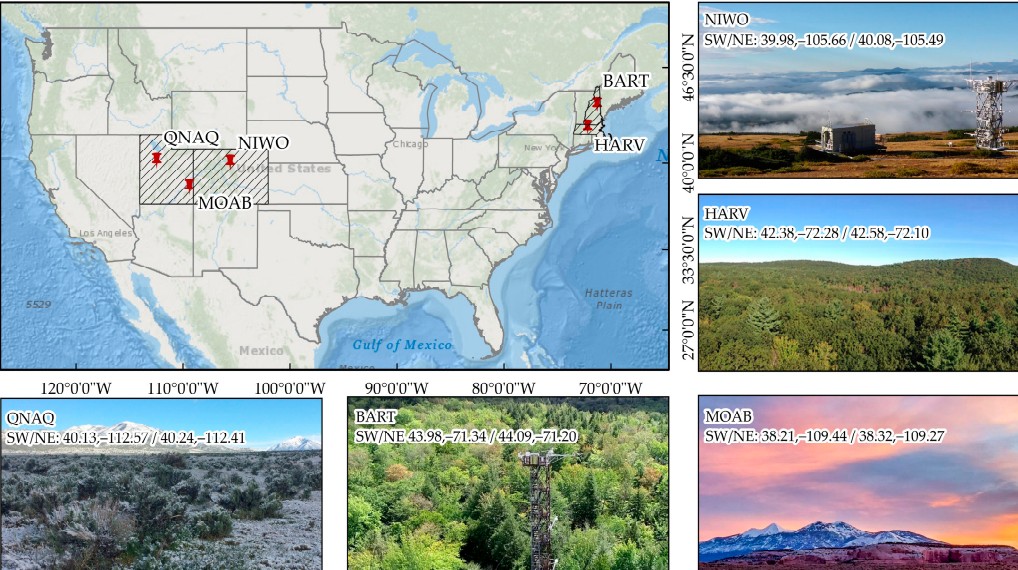

**Figure 2.** Schematic diagram of the location distribution of the study area.

**Table 1.** Detailed information of the experimental data [1]. FVC was categorized into classes I, II, III, and IV, which varied from 0 to 20%, 20% to 45%, 45% to 75% and 75% to 100%, respectively, and terrain slope was categorized into classes I, II, III and IV, which varied from 0 to 5°, 5° to 15°, 15° to 25° and 25° to 60°, respectively.

| Number | NEON Site Name | ICESat-2/ATL03 Time | FVC | Number | NEON Site Name | ICESat-2/ATL03 Time | Slope |
|--------|----------------|---------------------|-----|--------|----------------|---------------------|-------|
| Data1 | MOAB | 2022.07.13.Daytime | I | Data9 | MOAB | 2022.07.13.Daytime | I |
| Data2 | MOAB | 2020.09.21.Night | I | Data10 | MOAB | 2020.09.21.Night | I |
| Data3 | ONAQ | 2022.07.31.Daytime | II | Data11 | MOAB | 2021.09.16.Daytime | II |
| Data4 | ONAQ | 2022.07.31.Night | II | Data12 | MOAB | 2020.04.17.Night | II |
| Data5 | BART | 2020.07.03.Daytime | III | Data13 | NIWO | 2021.08.27.Daytime | III |
| Data6 | BART | 2019.09.03.Night | III | Data14 | NIWO | 2020.06.02.Night | III |
| Data7 | HARV | 2022.07.07.Daytime | IV | Data15 | NIWO | 2021.08.27.Daytime | IV |
| Data8 | HARV | 2020.08.09.Night | IV | Data16 | NIWO | 2020.06.02.Night | IV |

[1] To ensure the validity of the experiment, the slopes of Data1~Data8 are of the same classification, and the vegetation cover of Data9~Data16 are of the same classification.

*2.2. Methods*

2.2.1. The DRAGANN Algorithm

The DRAGANN algorithm is the denoising algorithm adopted by the ICESat-2/AIL08 product. There are two main steps. First, the data of the segment to be processed construct a KD-tree spatial index to find the domain points, which can improve the speed and efficiency of data processing. The total number of points within the circular domain of each photon is calculated by radius $R$ search, and a histogram of the neighborhood count distribution is generated. The signal photons are more densely distributed in space than the noise photons; therefore, the histogram generally shows a "bimodal" feature. The computational model is as follows:

$$R = \sqrt{\frac{P}{N_{Total} \cdot \pi}} \tag{1}$$

where $P$ is the number of valid neighborhood points in the search area, which is empirically set to 20, and $N_{Total}$ is the total number of photon point clouds.

Next, the Savitzky–Golay filtering algorithm is used to smooth the histogram to generate a smooth curve with an approximate "bimodal" distribution, and then the curve is peaked and fitted with a Gaussian function. The noise and signal Gaussian functions are determined with Gaussian fitting (Equation (2)). The intersection of the Gaussian function of the noise and the Gaussian function of the signal is the threshold. Points greater than this threshold are marked as signals, and points less than this threshold are marked as noise:

$$G(x) = \sum_{i=1}^{2} \alpha_i e^{-\frac{(x-\mu_i)^2}{2\sigma_i^2}} \tag{2}$$

where $\alpha_i$, $\sigma_i$ and $\mu_i$ are the amplitude, standard deviation and mean value of the Gaussian function, respectively.

2.2.2. The RBF Algorithm

The core idea of the RBF algorithm is to first count the distribution characteristics of the global photon point-cloud, eliminate the isolated point-cloud noise points and initially locate them to obtain the approximate range. Second, a Gaussian radial basis function is used locally for quadratic denoising. The specific steps are as follows.

First, coarse denoising is performed. The global photon point-cloud data in the area to be processed are subjected to elevation statistics (divided into 100 elevation slices); a

histogram of elevation distribution is obtained, and the histogram is smoothed using a five-point moving average algorithm:

$$s_{i,filt} = \alpha_1 s_{i-2} + \alpha_2 s_{i-1} + \alpha_3 s_i + \alpha_4 s_{i+1} + \alpha_5 s_{i+2} \tag{3}$$

where $\sum_{j=1,5} \alpha_j = 1$, $\alpha_1 = \alpha_5$, $\alpha_2 = \alpha_4$, and $\alpha = (\alpha_1, \alpha_2, \alpha_3, \alpha_4, \alpha_5) = (0.0625, 0.25, 0.375, 0.25, 0.0625)$.

Then, the ground and canopy elevation centers are determined with iterations. In the first iteration, the width of the elevation slice $g_1 = 1$, and all local maxima in the histogram are found and recorded in an index list. Then, the width of the elevation slice is changed continuously (Equation (4)), and if a certain maximum appears in the set of both old and new maxima, it is put into the new set. The two maxima are the ground and canopy elevation centers, respectively, until only two maxima remain and they meet the group distance of at least 8 histograms apart. This gives the approximate elevation range where the signal photons are located:

$$g_n = g_{n-1} + 1 \tag{4}$$

Next, secondary denoising is performed. The weighted cumulative distance sum of all adjacent points within its 15 m radius is calculated using the radial basis function for the coarse denoised points to obtain the density value $f_d(c)$, and a histogram of local density values is generated. The calculation equation is as follows:

$$\varnothing(r) = e^{-\left(\frac{r}{\sqrt{2}\sigma}\right)^2} \tag{5}$$

$$f_d(c) = \sum_{x \in D_c} \varnothing(||x - c||_a) \tag{6}$$

where $D_c = \{x \in D : ||x - c||_2 \leq 15 \text{ m}\}$, $||x - c||_2$ denotes the L2 parametrization, which corresponds to the Euclidean distance. The domain radius is chosen to be 15 m because the reflection radius of the feature is usually less than 15 m.

After smoothing the density value histogram, the maximum histogram is identified as $H_{max}(d_m)$, where the density value corresponding to the maximum histogram of $d_m$ is defined as the noise threshold, and the points with a density value less than the threshold $d_m$ are considered noise points and are eliminated. To eliminate the possible existence of high-density value noise photons, the density value is calculated again for the remaining points.

### 2.2.3. The DBSCAN Algorithm

The DBSCAN algorithm is a density-based spatial clustering algorithm that defines a class as the largest set of densely connected points, and clustering is accomplished by continuously searching for the largest set in the sample space. The photons in the maximum set are signal photons, and those not in the set are noise photons. The specific steps are as follows.

First, two parameters of the DBSCAN algorithm, Eps (domain radius) and *Minpts* (minimum number of inclusion points), are determined, and the determination of the parameters is particularly important. To ensure the validity of the comparison with other algorithms, the method of parameter selection is harmonized. In this study, the selection is made with reference to Zhang's method [31,32]. For simplicity, always use Eps =10 so that only *MinPts* need to be modified:

$$S = \delta s \cdot \delta h \tag{7}$$

where $S$ is the total area, $\delta s$ is the distance along the rail and $\delta h$ the elevation range.

For an ellipse with dist($p$, $q$) = Eps, its area s1 is as follows:

$$\text{s1} = \pi \cdot \text{Eps}^2 \tag{8}$$

$$\rho = N_{total} / S \cdot \text{s1} \tag{9}$$

where $\rho$ is the average density inside the circle and $N_{total}$ is the total number of photon point clouds.

Multiple experimental data were sampled; the above steps were repeated and then averaged for a better estimate of $\rho$, resulting in *Minpts* = 15:

$$Minpts \geq \rho \tag{10}$$

Next, each photon in the segment to be processed is traversed, and, when its point density within the domain radius exceeds *Minpts*, the photon is identified as a signal photon; otherwise, the photon is a noise photon.

### 2.2.4. Accuracy Verification

To effectively evaluate the adaptability of the three denoising algorithms, airborne LiDAR data were selected as the standard validation data, and recall $R$, accuracy $P$ and comprehensive evaluation index $F$ were used as the accuracy evaluation index [33]. The calculation model is as follows:

$$R = \frac{T_P}{T_P + F_N} \tag{11}$$

$$P = \frac{T_P}{T_P + F_p} \tag{12}$$

$$F = \frac{2PR}{P + R} \tag{13}$$

where $T_P$ denotes the number of photons that are actually signal points and are classified as signal points, $F_P$ denotes the number of photons that are actually noise but are classified as signal points and $F_N$ denotes the number of photons that are actually signal points but are classified as noise points.

## 3. Results

### 3.1. Analysis of the Effect of FVC on the Denoising Results of Three Algorithms

Using the above method, the denoising accuracy indices R, P and F for the three algorithms can be calculated for different FVCs (Table 2). From the table, the maximum F values of the DRAGANN algorithm are 0.916 and 0.948 for daytime and night-time, respectively, and the maximum F values of the DBSCAN algorithm are 0.913 and 0.953 for daytime and night-time, respectively, indicating that the performance of the DRAGANN algorithm and DBSCAN algorithm in denoising evaluation indices is optimal for vegetation cover Level III, i.e., 45–75%, in both daytime and night-time. The maximum F values of the RBF algorithm are 0.963 and 0.966 in daytime and night-time, respectively, indicating that the performance of the denoising evaluation indices is optimal at vegetation cover Level I. The maximum F values of the RBF algorithm are 0.963 and 0.966 in daytime and night-time, respectively.

**Table 2.** Evaluation indices of denoising results of Data1–Data8.

| Number | DRAGANN Algorithm | | | RBF Algorithm | | | DBSCAN Algorithm | | |
|---|---|---|---|---|---|---|---|---|---|
| | R | P | F | R | P | F | R | P | F |
| Data1 | 1 | 0.795 | 0.886 | 0.999 | 0.930 | 0.963 | 1 | 0.730 | 0.844 |
| Data2 | 1 | 0.832 | 0.908 | 1 | 0.934 | 0.966 | 0.998 | 0.840 | 0.952 |
| Data3 | 1 | 0.799 | 0.888 | 1 | 0.921 | 0.959 | 1 | 0.726 | 0.841 |
| Data4 | 1 | 0.859 | 0.924 | 1 | 0.925 | 0.961 | 1 | 0.859 | 0.924 |
| Data5 | 0.944 | 0.890 | 0.916 | 0.942 | 0.897 | 0.919 | 0.941 | 0.887 | 0.913 |
| Data6 | 1 | 0.901 | 0.948 | 0.998 | 0.903 | 0.948 | 0.998 | 0.912 | 0.953 |
| Data7 | 0.928 | 0.824 | 0.873 | 0.976 | 0.872 | 0.921 | 0.925 | 0.798 | 0.857 |
| Data8 | 0.982 | 0.874 | 0.925 | 0.998 | 0.857 | 0.922 | 0.897 | 0.887 | 0.892 |

The F value differences of Data1–Data8 during daytime and night-time were calculated, and the results are shown in Table 3. The difference in the F value of the RBF algorithm is the smallest, which is 0.016 and 0.015, respectively, and the difference in the DBSCAN algorithm is the largest, which is 0.044 and 0.039, respectively. It shows that FVC has the least influence on the denoising effect of the RBF algorithm and the greatest influence on the denoising effect of the DBSCAN algorithm.

**Table 3.** Difference in F values of different vegetation cover.

| Environmental Factors | DRAGANN Algorithm | | RBF Algorithm | | DBSCAN Algorithm | |
|---|---|---|---|---|---|---|
| | Daytime | Night | Daytime | Night | Daytime | Night |
| FVC | 0.024 | 0.021 | 0.016 | 0.015 | 0.044 | 0.039 |

To better analyze the influence of vegetation cover on the denoising effect of the three algorithms, a set of data with large differences in the denoising effects among the algorithms was selected for visual analysis. The denoising results for the area with Class III FVC and Class I slope are shown in Figure 3, indicating that under the conditions of high vegetation cover and flat terrain, all three algorithms can retain a large number of canopy signal photons. In the daytime, the denoising effect of the three algorithms is approximately the same; there is a small amount of missing signal photons under the canopy layer, and the phenomenon of continuous ground cannot be obtained. At night, the DRAGANN algorithm has a small number of unremoved noise photons above the canopy, the RBF algorithm has a relatively poor denoising effect, the signal photons below the canopy are missing and there are missing signal photons in the canopy. The DBSCAN algorithm has a better denoising effect, which can retain the signal photons in the canopy and get a continuous ground. Combined with Table 2, the visualization results are consistent with this observation. The F-value of the three algorithms does not differ much during the daytime, and the F-value of the DRAGANN algorithm and the RBF algorithm at night is 0.948, which is smaller than that of the DBSCAN algorithm.

*3.2. Analysis of the Effect of Slope on the Denoising Results of Three Algorithms*

Using the above method, the denoising accuracy indices R, P and F of the three algorithms can be calculated for different slopes (Table 4 and Figure 4). The recall R of both DRAGANN and DBSCAN algorithms is almost 1. Both algorithms have a higher probability of classifying the signal correctly as a signal, indicating that the slope has less influence on both algorithms.

**Table 4.** Evaluation indices of denoising results of Data9–Data16.

| Number | DRAGANN Algorithm | | | RBF Algorithm | | | DBSCAN Algorithm | | |
|---|---|---|---|---|---|---|---|---|---|
| | R | P | F | R | P | F | R | P | F |
| Data9 | 1 | 0.795 | 0.886 | 0.999 | 0.930 | 0.963 | 1 | 0.730 | 0.844 |
| Data10 | 1 | 0.832 | 0.908 | 1 | 0.934 | 0.966 | 0.998 | 0.840 | 0.912 |
| Data11 | 1 | 0.783 | 0.878 | 0.978 | 0.912 | 0.944 | 1 | 0.723 | 0.839 |
| Data12 | 1 | 0.826 | 0.905 | 0.988 | 0.909 | 0.947 | 1 | 0.830 | 0.907 |
| Data13 | 1 | 0.757 | 0.862 | 0.961 | 0.861 | 0.908 | 1 | 0.715 | 0.834 |
| Data14 | 1 | 0.813 | 0.897 | 0.981 | 0.850 | 0.911 | 1 | 0.821 | 0.902 |
| Data15 | 1 | 0.751 | 0.858 | 0.953 | 0.809 | 0.875 | 1 | 0.704 | 0.826 |
| Data16 | 1 | 0.807 | 0.893 | 0.959 | 0.811 | 0.879 | 1 | 0.817 | 0.899 |

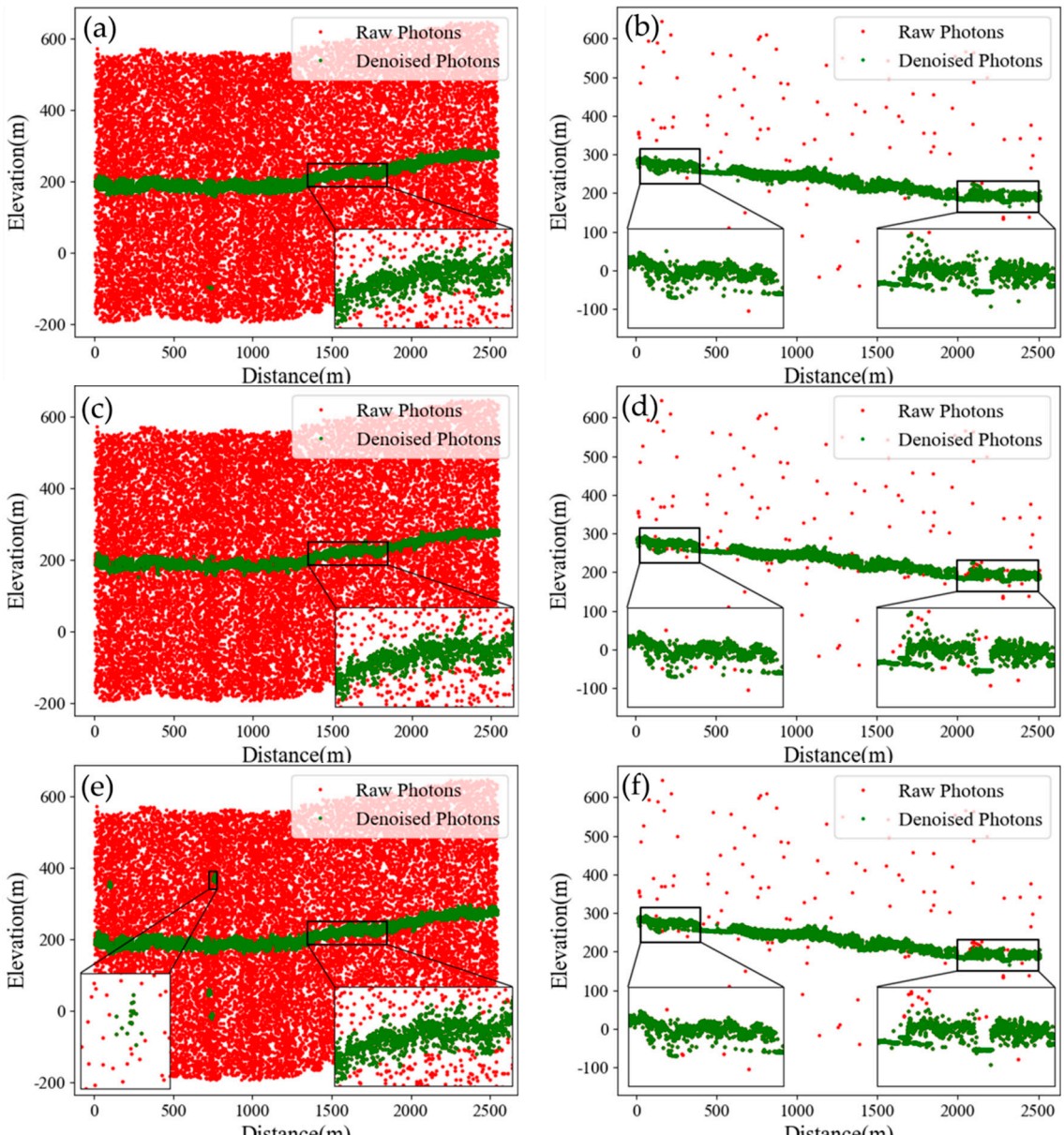

**Figure 3.** Point-cloud denoising results for high FVC. (**a**,**b**) are the daytime and night-time denoising results of the DRAGANN algorithm, respectively. (**c**,**d**) are the daytime and night-time denoising results of the RBF algorithm, respectively. (**e**,**f**) are the daytime and night-time denoising results of the DBSCAN algorithm, respectively.

As can be seen from Figure 4, all three algorithms have the best performance in denoising evaluation indexes when the terrain slope is the smallest, and the denoising effect deteriorates as the slope increases. The F value of the RBF algorithm varies greatly with the increase of slope level, and the F value of the DRAGANN algorithm and the DBSCAN algorithm varies very little, which indicates that terrain slope has a greater impact on the denoising effect of the RBF algorithm and has a similar degree of influence on the DRAGANN algorithm and DBSCAN algorithm, which are both smaller.

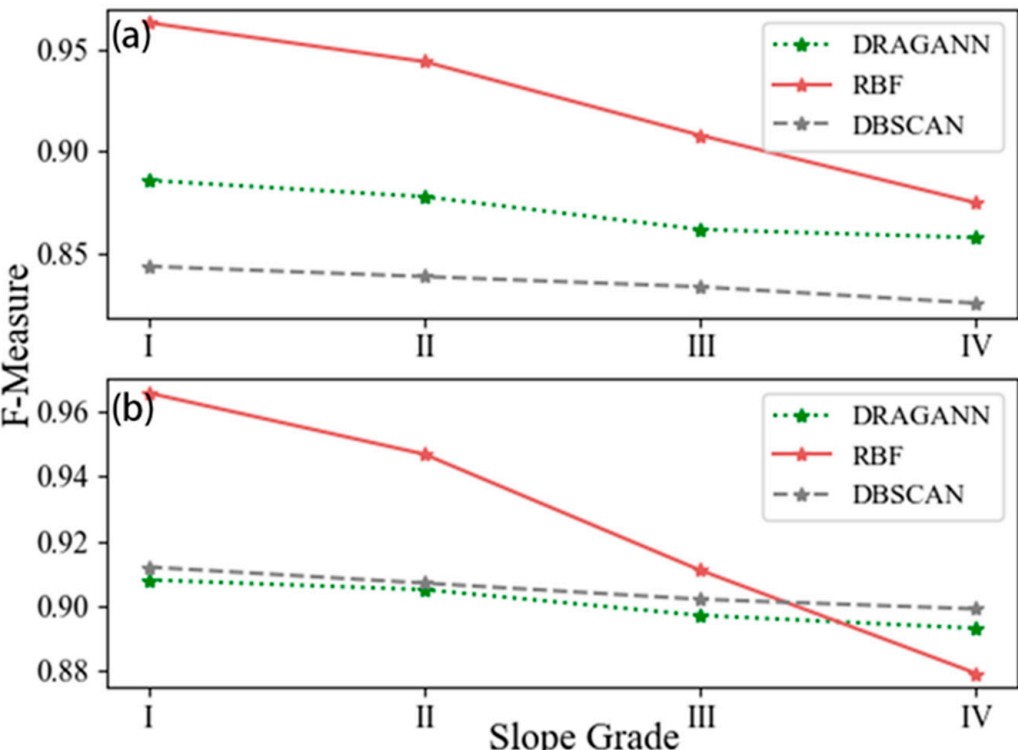

**Figure 4.** The comprehensive evaluation index F value of different slope grades. (**a**) The F value of daytime DRAGANN algorithm, RBF algorithm and DBSCAN algorithm at different slope grades. (**b**) The F value of night-time DRAGANN algorithm, RBF algorithm and DBSCAN algorithm at different slope grades.

Since the denoising effect of all three algorithms deteriorates with increasing slope, the high-slope area is selected for visual analysis. Among them, the denoising results for the area with FVC of Class I and slope of Class IV are shown in Figure 5, which shows that all three algorithms can remove a large number of noise photons under conditions of low FVC and large topographic undulations. Figure 5a,b,e,f show that although both the DRAGANN algorithm and the DBSCAN algorithm fail to completely remove the noise photons below the ground, their denoising effect is less influenced by the slope, and there are fewer isolated noise points at night; therefore, the denoising effect is better. Figure 5c,d show that, regardless of daytime or night-time, the denoising effect of the RBF algorithm is more affected by slope, and the signal photons are missing at the rumble-like slope and fail to obtain continuous ground. Combining Table 4 and Figure 5, in daytime, the F value of the DRAGANN algorithm and the DBSCAN algorithm are 0.858 and 0.826, respectively, which are smaller than the F value of the RBF algorithm, 0.875, which is because, although the denoising effect of the DRAGANN algorithm and the DBSCAN algorithm is less affected by slope, it is more affected by the solar background photons. At night, the solar background photons are less, and the F value of the DRAGANN algorithm and the DBSCAN algorithm are 0.893 and 0.899, respectively, which are greater than the F value of the RBF algorithm, 0.879. It can be seen that the results of Figure 5 and Table 4 are consistent.

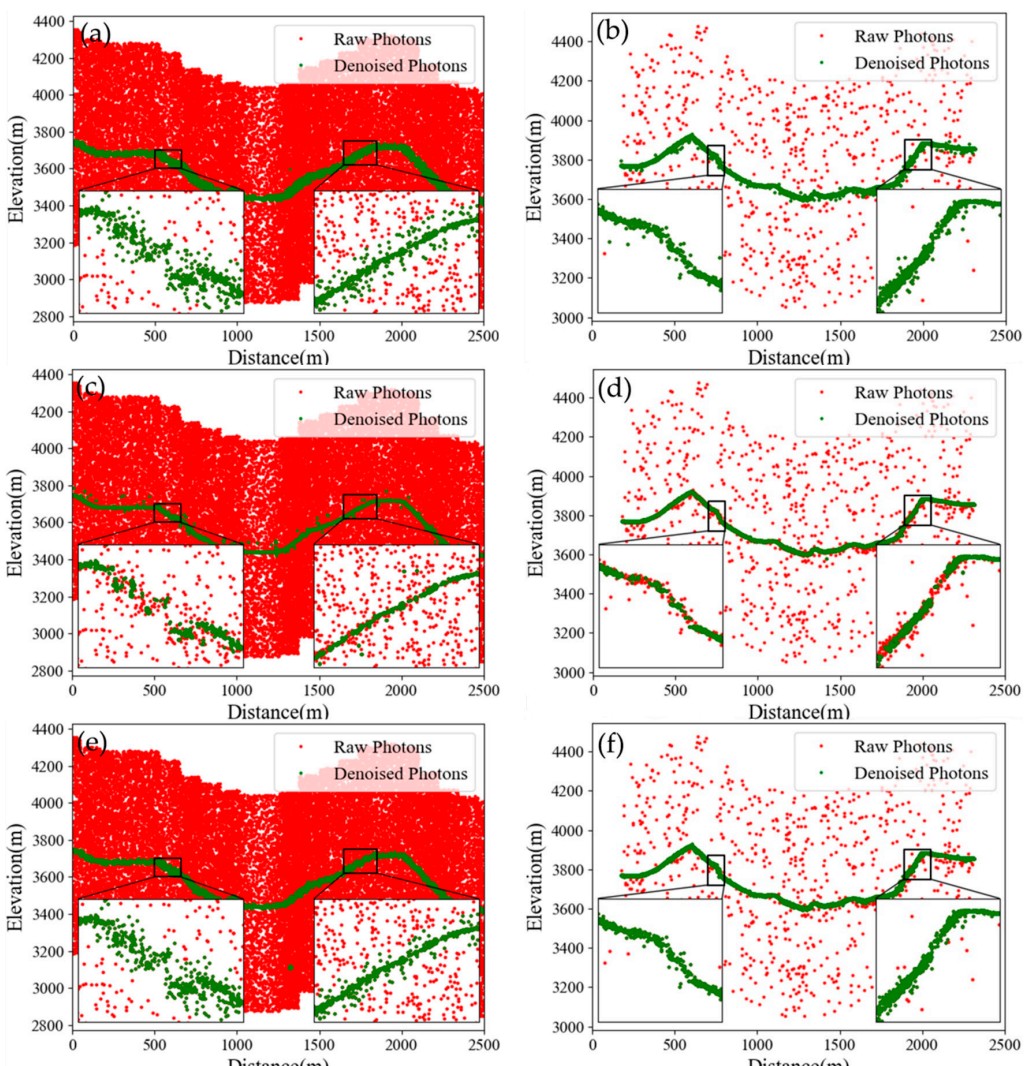

**Figure 5.** Point−cloud denoising results for high slope. (**a**,**b**) are the daytime and night-time denoising results of the DRAGANN algorithm, respectively. (**c**,**d**) are the daytime and night-time denoising results of the RBF algorithm, respectively. (**e**,**f**) are the daytime and night-time denoising results of the DBSCAN algorithm, respectively.

### 3.3. Analysis of the Effect of Observation Time on the Denoising Results of Three Algorithms

The average F values of Data1–Data16 in daytime and night-time are calculated separately (Table 5). From Table 5, the F values of the DRAGANN algorithm, RBF algorithm and DBSCAN algorithm are 0.880, 0.927 and 0.850 in the daytime and 0.914, 0.933 and 0.930 in the night-time, respectively. The average denoising effect of all three algorithms is better in the night-time than in the daytime, which indicates that the observation time, i.e., solar noise photons, has an effect on all three algorithms. Among them, the DBSCAN algorithm has the smallest F value during the daytime, indicating that it has the greatest influence on the denoising effect of the DBSCAN algorithm. In addition, from an overall perspective, the RBF algorithm has a higher mean value of F compared to the other two algorithms, indicating that its denoising effect is better overall.

Figure 6 shows the denoising results of the area with FVC and slope Level I at different time periods, indicating that all three algorithms can remove a large number of noise photons to obtain continuous terrain at different time periods. Figure 6a,b show that the denoising effect of the DRAGANN algorithm in daytime and night-time is approximately the same, and both fail to remove the noise photons above and below the ground. Figure 6c,d show that the RBF algorithm has a better denoising effect without an obvious misclassifica-

tion phenomenon. Figure 6e,f show that the DBSCAN algorithm fails to remove the noise photons below the ground during the daytime and misclassifies multiple clusters of noise photons into signal photons, and a small number of noise photons are also retained at night. Combined with Table 2 in the previous section, the RBF algorithm has the highest F value in both daytime and night-time, which are 0.963 and 0.966, respectively; the F value of the DRAGANN algorithm in daytime is 0.886 higher than that of the DBSCAN algorithm in daytime which is 0.844, and the F value of the DRAGANN algorithm in night-time is 0.908 lower than that in the DBSCAN algorithm at night-time which is 0.952. It is highly consistent with the visualization results.

**Table 5.** Average values of F values for different time periods.

| Time | DRAGANN Algorithm | RBF Algorithm | DBSCAN Algorithm |
|---|---|---|---|
| Daytime | 0.880 | 0.927 | 0.850 |
| Night | 0.914 | 0.933 | 0.930 |

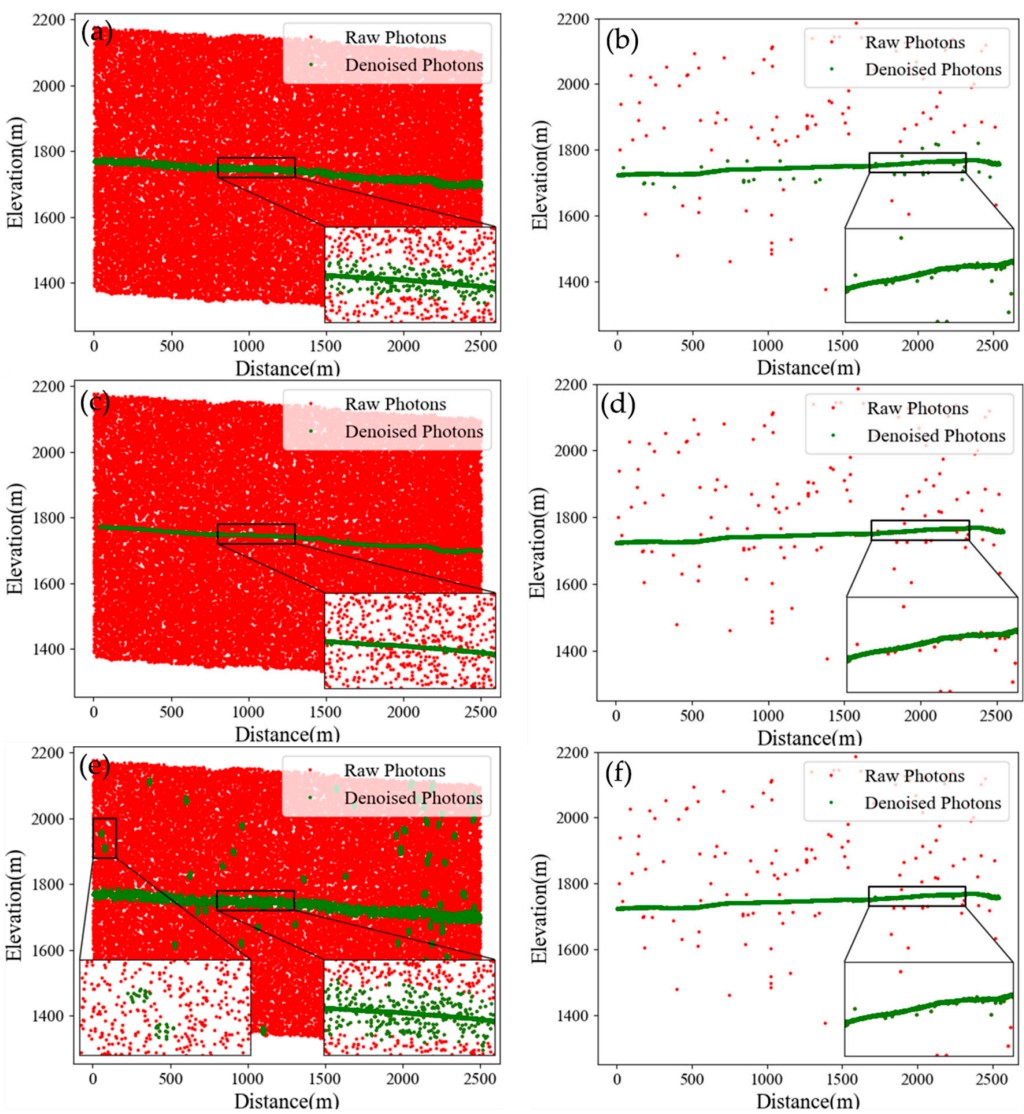

**Figure 6.** Point-cloud denoising results under different time period conditions. (**a**,**b**) are the daytime and night-time denoising results of the DRAGANN algorithm, respectively. (**c**,**d**) are the daytime and night-time denoising results of the RBF algorithm, respectively. (**e**,**f**) are the daytime and night-time denoising results of the DBSCAN algorithm, respectively.

## 4. Discussion

The DRAGANN algorithm local density is a fixed circle for domain search statistics, the parameter P required to determine the radius of the circle relies only on empirical values and the density histogram may not find the ideal "noise + signal" bimodal distribution due to the differences in spatial characteristics of point clouds in different terrains with different signal-to-noise ratios [24,34].

In daytime, the signal photons are sparse, the signal-to-noise ratio is low and the density histogram is characterized by noise on the left, high and narrow, and signal on the right, low and wide, when the noise and signal peaks are easily mixed (Figure 7a), and the threshold error extracted is larger. At night, the signal-to-noise ratio is higher, and the density histogram generally shows the characteristics of noise on the left, high and narrow, and signal on the right, low and wide. When the FVC is less than 45% or more than 75%, the noise peaks and signal peaks are also easy to mix (Figure 7b). When the FVC is 45~75%, the noise peaks and signal peaks generally do not intersect (Figure 7c), and the extracted thresholds easily achieve signal photon separation. Therefore, the DRAGANN algorithm has a better denoising effect at night and when the FVC is 45~75%. Since the parameter P required by the DRAGANN algorithm to determine the radius of the circle is a denoising algorithm based on the density characteristics of the global photon point cloud, the slope has less effect on the signal-to-noise ratio of the global photon point cloud; therefore, the slope has less effect on the denoising effect of the DRAGANN algorithm.

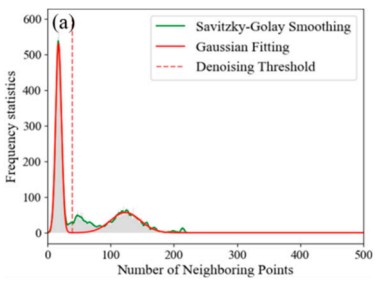 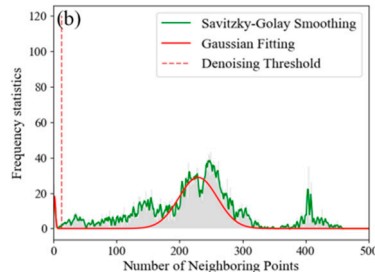 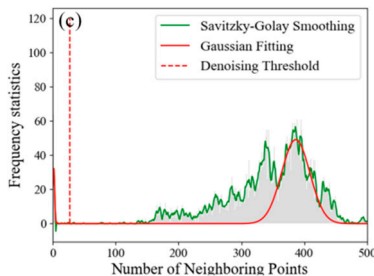

**Figure 7.** Threshold extraction results. (**a**) Daytime threshold extraction results. (**b**) Threshold extraction results when the FVC is 45% or more than 75% at night. (**c**) Threshold extraction results at night when the FVC is 45~75%.

The RBF algorithm first processes the global photon point cloud based on its density characteristics to obtain the approximate range of the signal photon point cloud. Second, based on the density characteristics of the local photon point cloud, the photon point-cloud neighborhood is searched with the Gaussian radial basis function, which takes into account the anisotropy of the point-cloud distribution characteristics, i.e., more extensions in the horizontal direction than in the vertical direction, and has a certain universality relative to circles; therefore, the denoising effect of the RBF algorithm is better than that of the DRAGANN algorithm and the DBSCAN algorithm from the overall perspective.

However, its algorithm performance cannot be adapted to all types of areas. When the slope of the terrain increases or the FVC increases, the extension of the point cloud in the vertical direction increases even more than that in the horizontal direction. Therefore, the denoising effect of the RBF algorithm becomes worse with increasing FVC and worse with increasing slope, and it is influenced by the slope. The threshold value of the algorithm is uncertain and closely related to the shooting conditions, point-cloud density, processing range and feature category [35], and the selection of the threshold value and the improvement of its self-adaptability are the key factors affecting the denoising effect of the algorithm.

The RBF algorithm adopts a "global–local" approach, and the observation time of the photon point cloud has less effect on the signal-to-noise ratio of the local photon point cloud; therefore, the observation time has less effect on the denoising effect of the RBF algorithm.

The DBSCAN algorithm, as a density clustering-based algorithm, can discover the maximum set of arbitrary shapes through the connectivity of density among photon point clouds to effectively identify noise and signal points [36]. Therefore, the DBSCAN algorithm is less effective in clustering outlier "photon clusters" that match the parameters; when the daytime noisy photon point cloud is dense and matches the parameters, it is easily misclassified as signal photons, and therefore, the algorithm is affected by the observation time.

In addition, the DBSCAN algorithm is sensitive to the parameters *Eps* (field radius) and *Minpts* (field points) [23], and it cannot be widely applied to all areas by manually selecting the parameters. When the vegetation cover is lower than 45%, the canopy signal photons are sparse, even when the ground condition is close to bare ground and the noise photons are similar to the ground photon density; then, they are misclassified. When the vegetation cover is 45~75%, the noise photons are significantly different from the canopy photon and ground photon density and can be correctly classified. When the vegetation cover is higher than 75%, the ground photon density is lower than the canopy photon density. It is easy to mistakenly reject ground photons as noise photons and cannot obtain continuous ground. The parameters in the experiment are selected based on the global photon point-cloud density feature [31,32]; therefore, the slope has little effect on the denoising effect of the DBSCAN algorithm.

## 5. Conclusions

To investigate the adaptability of different denoising algorithms for photon-counting LiDAR data under different spatial and temporal conditions and their algorithmic mechanisms, the DRAGANN algorithm, RBF algorithm and DBSCAN algorithm are selected to carry out denoising experiments and accuracy analysis for ICESat-2 data under different spatial and temporal conditions, and the following conclusions are made.

The DRAGANN algorithm, RBF algorithm and DBSCAN algorithm have different adaptabilities under different conditions. (1) The DRAGANN algorithm is mainly influenced by parameter selection and threshold selection and is only applicable to night-time and moderate daytime FVC (45~75%) data. (2) The RBF algorithm uses a Gaussian radial basis function to count the local density, and the weight in the horizontal direction is greater than that in the vertical direction, which makes the denoising accuracy lower with increasing FVC and lower with increasing slope; therefore, it is applicable for all-day data with low terrain slope (0~5°) and low FVC (0~20%). (3) The DBSCAN algorithm is mainly influenced by the parameter selection and the defects of the algorithm clustering and is applicable to the data with moderate FVC (45~75%) at night, which is consistent with the adaptability of the DRAGANN algorithm, with the difference that the DBSCAN algorithm is more influenced by the time period, but the night-time denoising accuracy is relatively high. The DRAGANN algorithm and DBSCAN algorithm are affected by FVC and time period, and the parameters can be optimized by establishing the relationship between parameters and the point-cloud signal-to-noise ratio; the RBF algorithm is affected by slope, and the development of an adaptive algorithm along the photon point-cloud direction is considered.

Compared with airborne LIDAR, the trajectory of ICESat-2 is longer, the coverage density is higher, the data volume is larger and the algorithm for denoising its data is more demanding. Therefore, the adaptive study of the denoising algorithm for ICESat-2 data has some reference significance for the processing of photon-counting LIDAR data by later generations and provides scientific support for the application of photon-counting LIDAR data in various fields.

**Author Contributions:** Conceptualization, M.K. and Y.X.; methodology, M.K., Y.X. and J.W.; validation, M.K. and Y.X.; software, M.K. and J.W.; data curation, J.W., F.C. and Y.X.; writing—original draft preparation, M.K., F.C. and J.W.; writing—review and editing, M.K. and F.C. All authors have read and agreed to the published version of the manuscript.

**Funding:** This research was supported in part by the National Natural Science Foundation of China under No.41961060, in part by the Yunnan Fundamental Research Projects (No.202001AU070060/ No.202101AT070078), in part by the Technology Talent and Platform Project of Yunnan Province of China for the "Remote sensing estimation of regional vegetation carbon storage" [No. 202305AO350003], and in part by the Geology and Mineral Resources Exploration Development Bureau of Yunnan Province Science and Technology Innovation Project under No. 202235.

**Data Availability Statement:** ICESat-2/ATLAS LiDAR data used in this study are openly available at https://nsidc.org/data/icesat-2/data-sets (accessed on 6 October 2022); the airborne data used in this study are openly available on the U.S. National Ecological Observatory Network https://www.neonscience.org/data-collection/lidar (accessed on 6 October 2022).

**Conflicts of Interest:** The authors declare no conflict of interest.

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
