# Peer review of "Research on the Adaptability of Typical Denoising Algorithms Based on ICESat-2 Data"

_remotesensing, doi:10.3390/rs15153884_

Round 1

Reviewer 1 Report

This study compared the adaptability of three photonic point cloud denoising algorithms, DRAGANN, RBF and DBSCAN, under different spatial and temporal conditions, and compared the reasons for the different adaptability of different algorithms. However, there are some writing problems that need to be clarified. In addition, some specific problems and comments are seen as the following.

1Lines 203-204, the DBSCAN clustering method is very sensitive to its two key parameters, and different parameters will bring very different results. The author does not discuss the parameters here, only using the previous experience value, which is not rigorous enough. It is suggested that the author explain the advantages of the previous research on the selection of these two parameters, so as to further explain the scientific nature of your choice.

2Lines 246-252, is this the same as the F value ? It is suggested that the author supplement the explanation and echo before and after. Similar problems also exist in lines 287-298 and 320-327.

3Lines 272-275, it is suggested that the author replace Table 4 with a line chart, which will be more intuitive.

4Lines 302-305, the name of the figure is inconsistent with the serial number of the upper left corner of the figure, please correct it.

5Line 328, the fourth subfigure (b) of Figure 4 should be changed to (d).

6There are many denoising algorithms now, why choose these three methods for comparison ?

7What is the innovation and significance of the article ?

8Reference 21 should be“A noise removal algorithm based on OPTICS for photon-counting LiDAR data”.

9Misspelled author name in Reference 35.

Minor editing of English language required

Reviewer 2 Report

This paper presents an interesting study on the application of different denoising algorithms for photon-counting LIDAR data under different spatial and temporal conditions. As a test case, the authors use the ICESat-2 data. To some extent, the paper presents novel material as the pipeline of work is new, even if it can be considered a technical implementation and not, in a strict sense, a scientific article. This paper is a useful contribution and provides ideas that are significant for LIDAR data processing. The paper is mostly well-written, but I have some comments. In the Introduction, the authors listed many studies without fully describing what the limitations of the studies are and how they will overcome these limitations. It might be worth to perform comparisons between the different methodologies to identify the knowledge gap and how this paper addresses it. I suggest that the authors summarize recent developments in this field, highlight what needs further research and development, and why the existing studies are insufficient. It should be explained in more detail why the ATL03 is used to conduct the experiments.
